# Valence-Driven Cognitive Flexibility: Neurochemical and Circuit-Level Insights from Animal Models and Their Relevance to Schizophrenia

**DOI:** 10.3390/biom15081154

**Published:** 2025-08-11

**Authors:** Kfir Asraf, Inna Gaisler-Salomon

**Affiliations:** 1School of Psychological Sciences, University of Haifa, Haifa 3498838, Israel; kasraf01@campus.haifa.ac.il; 2Integrative Brain and Behavior Research Center, University of Haifa, Haifa 3498838, Israel

**Keywords:** reversal learning, attentional set shifting, intra-dimensional set shifting, extra-dimensional set shifting, cognitive deficits, animal models

## Abstract

Cognitive flexibility, the ability to adapt behavior to changing environmental demands, is a core deficit in schizophrenia (SZ), that predicts disease progression. This review synthesizes findings on the neural substates of cognitive flexibility by using a framework that distinguishes animal model tasks by their motivational valence: aversive versus appetitive. While human studies using tasks like the Wisconsin Card Sorting Test (WCST) reveal significant cognitive inflexibility in SZ, particularly in set shifting, rodent models provide important mechanistic insights. The current literature suggests that aversive tasks, such as water mazes, and appetitive tasks, such as the Birrel–Brown discrimination task, engage distinct neural circuits, despite assessing supposedly similar cognitive processes. Aversive paradigms primarily rely on hippocampal–medial prefrontal cortex (mPFC) pathways, whereas appetitive tasks heavily involve orbitofrontal cortex (OFC)–striatal circuits, with significant modulation by dopamine and serotonin. Both valences seem to require an intact balance of glutamate and GABA transmission within prefrontal regions. This framework helps clarify inconsistencies in the literature and underscores how motivational context shapes the neural substrates of cognitive flexibility.

## 1. Introduction

The ability to change behavior in line with changing environmental demands, known as cognitive flexibility, is deficient in many psychiatric and neurological disorders such as Parkinson’s disease [1], obsessive–compulsive disorder [2] and schizophrenia (SZ) [3]. In SZ, cognitive flexibility deficits are associated with other cognitive abnormalities that emerge in adolescence [4] and serve as a predisposing factor to psychosis and a predictor of disease progression [5]. Although cognitive abilities decline with age in both healthy individuals [6] and those with SZ [7], the decline in cognitive flexibility is more pronounced in SZ, pointing to underlying disease-specific processes in this particular cognitive capacity [8].

Cognitive flexibility is commonly assessed in both humans and animals using reversal learning and attentional set-shifting tasks. In reversal learning tasks, also called intra-dimensional (ID) shifting, subjects learn a stimulus–reward association that is subsequently reversed—e.g., a reward initially associated with the color red is now associated with the color green [9]. In attentional set shifting, or extra-dimensional set shifting (EDSS) tasks, subjects are required to shift their response across stimulus dimensions—e.g., from color to odor—posing a greater cognitive challenge than shifts within the same dimension [10].

Error types in reversal learning or attentional set shifting, in both human and animal studies, can reveal distinct patterns of cognitive inflexibility. Perseverative errors involve the repeated selection of a previously rewarded option before ever choosing the new correct one, while regressive errors occur when reverting back to the old choice after initially selecting the correct one. These error types are qualitatively distinct and may involve different neural mechanisms [11,12].

In this review, we present a comprehensive overview of tasks commonly used to assess intra- and extra-dimensional set shifting in humans, highlighting key findings from studies involving individuals with SZ. We then describe animal research—primarily from the last two decades—that enables a mechanistic investigation of cognitive flexibility capacities. A distinctive feature of our approach is the categorization of animal cognitive flexibility tasks according to their motivational valence—aversive or appetitive—a framework that has received limited attention in prior reviews. By distinguishing these task types, we aim to clarify how different motivational and emotional contexts engage distinct neural circuits. We then synthesize current evidence on the neural and neurochemical substrates underlying these capacities, emphasizing how neurotransmitter systems may differentially contribute to performance in aversive versus appetitive paradigms. Finally, we identify critical gaps in the literature and propose new directions for future research informed by this motivational dichotomy.

The review process was conducted between December 2024 and June 2025 and was performed across several academic databases, including Google Scholar and PubMed. The search pertaining to animal studies, which constituted the main focus of the review, was limited to articles published between 2000 and 2025. We searched for articles that captured three core domains: the behavioral model, brain region, and neurotransmitter. The search strings used aimed at capturing all three domains, based on knowledge that emerged from the initial search strategy (for example, “Cognitive flexibility AND Hippocampus AND Glutamate”). To reduce bias, multiple search string combinations were used, with different behavioral assays/models (i.e., Morris Water Maze/T-maze), brain regions (i.e., mPFC/Hippocampus) and neurotransmitters (i.e., Glutamate/GABA/Dopamine; see below).

Initial screening was conducted by reviewing titles and abstracts. Potentially relevant articles were then subjected to a full-text review. Studies were included if they (1) were peer-reviewed journal articles; (2) were published in English; (3) described a behavior cognitive flexibility assay in rats or mice. We then organized the studies based on valence, task type, implicated brain regions or circuits, and neurotransmitter systems. This approach was informed by prevailing patterns across diverse studies and methodologies. As detailed below, this revealed convergent findings from independent research groups using varied experimental paradigms, highlighting consistent themes rather than selective bias.

## 2. Assessing Cognitive Flexibility Deficits in SZ

The Wisconsin Card Sorting Test (WCST) is the most prevalent neurocognitive test for assessing EDSS in humans [13]. In this task, participants are presented with cards that vary along three dimensions, color, shape and number, and are required to deduce the correct sorting dimension through trial and error. After reaching a predefined correct trial criterion, the sorting dimension is changed without notice, requiring the participant to re-engage in trial-and-error learning to identify the new rule [13]. A closely related assessment is the more structured Intra-Dimensional/Extra-Dimensional (ID/ED) Set Shift Task, which is part of the commonly used Cambridge Neuropsychological Test Automated Battery (CANTAB) [3]. In both tasks, individuals with SZ perform comparably to healthy controls in stages requiring simple discrimination or ID shifting. However, they exhibit a higher error rate and lower success in reaching criterion in stages requiring ED shifting [14].

In a different task, the Trail Making Test, participants are first instructed to connect a sequence of numbered circles (1 to 25) as quickly as possible, assessing processing speed and visual search capabilities (TMT-A). In TMT-B, they are instructed to connect a sequence that alternates between numbers and letters (e.g., 1-A-2-B-3-C), requiring cognitive flexibility. Deficient performance on the TMT-B is observed in individuals with SZ and their unaffected siblings, suggesting that cognitive inflexibility may be an endophenotype of SZ and is not secondary to illness [15,16].

Cognitive flexibility in valuing reward has also been assessed, e.g., in tasks that require probabilistic reversal learning (PRL). Participants are initially trained to choose between two unevenly rewarded stimuli. Once they learn to favor the more frequently rewarded stimulus, the reward contingencies are reversed, making the previously less rewarded option the one now more likely to yield a reward. Patients with SZ are comparable to healthy controls in their ability to learn the initial discrimination contingencies but perform significantly worse when contingencies are reversed [17]. Additional tasks measuring cognitive flexibility in humans have been described and are largely similar to the tasks described above with some variations in stimulus type and response requirement [18,19].

Human studies link cognitive flexibility to the prefrontal cortex (PFC) and striatum, with glutamate and dopamine emerging as key neurotransmitters [14]. Such findings are inherently correlational; however, they provide important clues for understanding the brain circuitry involved in cognitive flexibility processes. For example, a polymorphism in the gene encoding Catechol-O-Methyltransferase (COMT), which catabolizes dopamine, was found to be associated with perseverative errors in the WCST [20]. Functional MRI studies further support a distributed network: one study linked OFC function to reversal learning and ventrolateral PFC activity with EDSS [21]. However, other research has found EDSS to be associated with activation of both the medial and lateral PFC [22]. Notably, fMRI tasks like the WCST involve overlapping processes (e.g., attention, inhibition), making it hard to isolate cognitive flexibility—an issue compounded by fMRI’s limited temporal resolution.

As mentioned above, human studies offer limited insight into causal mechanisms. Animal models, most commonly in mice and rats, offer a constantly expanding toolbox for manipulating specific genes, receptors, brain regions, cell populations and neural circuits. This level of experimental control provides insights into the biological basis of reversal learning and set shifting and allows researchers to establish causal links between synaptic mechanisms and cognitive flexibility.

## 3. Assessing Cognitive Flexibility Deficits in Rodents: Methodology, Neural Circuits and Neurotransmitters

Assessing cognitive flexibility in rodent models has been performed in the context of different pathologies, including SZ [23,24,25], autism [26], drug addiction [27], Alzheimer’s disease [28], and obsessive–compulsive disorder [29]. In these different contexts, aversive or appetitive learning paradigms have been used to assess reversal learning and set shifting. Below, we define the similarities and differences between these categories, describe some of the key tasks in each category (Figure 1) and examine the neural substrates involved.

### 3.1. Aversive Learning Tasks

Aversive tasks for measuring cognitive flexibility use negative reinforcement or punishment as a motivating factor. Commonly, aversive tasks in rodents are performed using water maze-based paradigms, e.g., the Morris Water Maze (MWM) [30] or water T-maze task [31], with platform finding as negative reinforcement. These tasks are ‘ecological’ in the sense that they leverage rodents’ innate swimming and spatial navigation abilities, rely on their intrinsic motivation to escape from water, and do not require behaviors beyond the animal’s natural repertoire. In both the MWM and water T-maze, rodents first learn the location of a hidden platform during the Acquisition stage. Once they reach a specific learning criterion, e.g., six consecutive correct trials, the platform is relocated to a new position, initiating the Reversal stage. Finally, in the EDSS stage, the animal must shift strategy and rely on non-spatial cues, e.g., a visual signal, to locate the platform. To reach a criterion in the EDSS stage, animals must inhibit their spatial search strategy and acquire a new strategy based on, e.g., visual cues [25,32].

Flexibility in aversive tasks can be extended to other paradigms requiring a contingency change; for example, fear conditioning tasks where footshock is the aversive reinforcer that elicits an innate fear response. For example, animals are first taught to discriminate between two different contexts, one predicting a safe environment and the other signaling an aversive stimulus; in such tasks, flexibility is defined as the ability to reverse the ‘safe’/’unsafe’ contingencies [33]. A variation in this task requires mice to shift from reward to punishment learning [34]. Cognitive flexibility is also required in latent inhibition tasks, where an animal is taught to ignore a stimulus that then becomes relevant in predicting an aversive event [35,36], and in fear extinction tasks, where a previously meaningful cue loses its significance [37]. However, these paradigms primarily assess learning abilities rather than cognitive flexibility and are therefore beyond the scope of this review.

### 3.2. Appetitive Learning Tasks

Appetitive tasks use positive reinforcement to drive behavior and rely on discrimination between sensory cues or modalities. Such tasks generally require a longer acquisition period compared to aversive tasks and usually necessitate food or water restriction to enhance motivation.

The Birrell–Brown attentional set-shifting task (ASST) involves a sequence of discrimination and reversal learning trials in which rodents use either the digging medium or odor cues to locate a food reward hidden in one of two bowls. Across task stages, the relevant stimulus dimension guiding correct choices changes. In the ID shift (reversal) phase, the rodent is required to choose a new cue within the same dimension (e.g., switching from one odor to another, regardless of the medium). In the EDSS phase, the rodent must shift attention to a different stimulus dimension altogether (e.g., from odor to digging medium), reflecting a higher demand on cognitive flexibility.

Other appetitive tasks assessing cognitive flexibility rely on visual cues and require either lever pressing or touchscreen activation. Such operant learning tasks commonly assess reversal but not EDSS, since the latter is more difficult to measure in visual cue-based tasks. For example, Dickson et al. [38] reported an exceptionally high error rate—between 400 and 1000—in the EDSS phase of a lever pressing task. Such outcomes may reflect not only cognitive flexibility deficits but also species-specific limitations in processing visual stimuli, making it difficult to isolate the cognitive component.

As an alternative approach, strategy-shifting tasks assess cognitive flexibility across different domains, reducing reliance on visual discrimination and allowing for a broader evaluation of shifting abilities. For example, a task introduced by Ragozzino et al. [39] requires the rodent to shift from a spatial strategy (‘reward is in the North arm’) to a response strategy (‘turn Right to obtain reward’) in a cross-maze apparatus with food reward. Similar studies have since been described, requiring shifting between different strategies (e.g., between a response and a visual cue strategy (‘reward is where the visual cue is’) [40]. These tasks enable testing both ID and ED shifts (e.g., response–response or place–response, respectively) and support a range of protocol variations, including shifts between visual, spatial, non-spatial visual, and non-spatial auditory tasks.

Appetitive tasks offer the advantage of precise control over stimuli and reinforcement. However, lever pressing and touchscreen-based tasks require rodents to learn behaviors that are not innate, necessitating acclimation, auto-shaping and pretraining before discrimination learning can begin. Furthermore, both lever pressing and digging-based tasks require food restriction; animals are usually kept at 80–85% of their body weight, which may alter the motivational state compared to non-restricted animals. These technical details prompt theoretical questions about the nature of reinforcement and the impact of motivational states on learning. For example, one may wonder whether food reward in food-restricted animals is indeed positive reinforcement or the removal of threat, playing the same role as the platform in water-maze tasks. Moreover, aversive tasks often evoke stress or fear responses, which can either impair or enhance cognitive flexibility depending on intensity and context. In contrast, appetitive tasks rely on reward-seeking behavior, which may engage different neural circuits and learning strategies, potentially influencing how flexibly an animal can shift between rules or contingencies.

### 3.3. The Neural Substrates of Cognitive Flexibility

While aversive and appetitive tasks presumably tap into the same cognitive capacity, differences in learning contingencies, paradigm specifics, and motivational states may recruit distinct neural circuits and neurotransmitter systems. Below, we outline the neural substrates of aversive and appetitive reversal and set-shifting tasks, aiming to highlight both their shared and distinct features (see Table 1).

Classical lesion studies and more recent chemogenetic investigations implicate three main brain structures in aversive, water maze-based tasks: the dorsal hippocampus, which is required for spatial learning [41,42,43], the caudate nucleus of the striatum which is associated with cue–response learning [44], and the medial PFC (mPFC), which plays a pivotal role in reversal learning and strategy switching [45,46]. Similar brain regions are also engaged in appetitive tasks, such as the Birrell–Brown attentional set-shifting task, though the specific contributions of PFC subregions differ depending on the cognitive demands. In this task, intact hippocampal function is required for reversal and EDSS learning; Marquis et al. [47] showed that juvenile rats with early-life ventral hippocampal lesions show intact acquisition but disrupted EDSS and late-phase reversal performance. Within the PFC, different subregions mediate distinct aspects of the task: lesions to the anterior cingulate impair both compound discrimination and reversal, increasing regressive and total errors [48]. OFC lesions produce selective deficits in reversal learning, whereas mPFC damage specifically disrupts EDSS [49].

Data from operant learning tasks reinforce this functional dissociation within the PFC. Boulougouris et al. [50] showed that lesions to the OFC or infralimbic or prelimbic mPFC impair the reversal of reward contingencies but leave the retention of previously acquired spatial discrimination intact. Further studies using an appetitive visual/spatial alternation task are in line with these findings and show that temporary OFC inactivation selectively disrupts reversal learning while leaving acquisition and EDSS unaffected [51].

Beyond the PFC, the dorsomedial striatum, which is innervated by both the mPFC and OFC, contributes to spatial and non-spatial appetitive reversal learning [52]. This region is thought to support the consistent execution of a selected strategy, while the inhibition of the previously relevant strategy or the generation of a new one is mainly attributed to the PFC. Another region found to play a role in cognitive flexibility is the locus coeruleus (LC); the optogenetic inhibition of this region—a major source of noradrenergic input to the mPFC [53]—impairs reversal learning and EDSS in the Birrel–Brown task [54]. Furthermore, the pharmacological deafferation of noradrenergic input in the mPFC selectively impairs EDSS [55], providing further support for the importance of LC-mPFC projections in cognitive flexibility.

#### 3.3.1. Glutamate

The most commonly investigated neurotransmitter in the context of cognitive flexibility is glutamate. Research has focused particularly on the NMDA receptor, which plays a role in basic learning processes [56] and is implicated in disorders associated with poor attentional set-shifting abilities such as SZ [57] and autism [58].

Pharmacological studies show that the acute administration of the non-competitive NMDA receptor blocker MK-801 impairs reversal learning at doses that leave acquisition learning intact. For example, Bardgett et al. [59] found that reversal is selectively impaired by MK-801, injected subcutaneously at 0.05 mg/kg to adolescent mice (PND 35–42) before both acquisition and reversal, in a water T-maze task; a higher dose (or 0.10 mg/kg) impairs both acquisition and reversal. A similar dose-dependent disruption of reversal was found in juvenile and peri-adolescent rats (PND 21, 26, or 30) tested in an appetitive T-maze task [60]. The acute administration of the NMDA receptor blocker PCP also induced selective reversal deficits in rats tested in an operant lever pressing task [61,62]. Interestingly, the optogenetic stimulation of parvalbumin (PV)-positive GABAergic interneurons in the PFC and ventral hippocampus rescued EDSS deficits following acute MK-801 [63], implying that attentional set shifting relies on an intact excitatory/inhibitory (E/I) balance in these regions.

Chronic NMDA receptor blockade also differentially affects acquisition and reversal learning. The administration of MK-801 (0.1–0.4 mg/kg) for several weeks in either juvenile [64] or adult rats [65] disrupted reversal learning but not spatial acquisition in the MWM. Interestingly, chronic MK-801 treatment also affected the quality of performance during reversal learning, leading to perseverative behaviors and inefficient spatial strategy use [65].

Not surprisingly, the intra-cortical administration of MK-801 selectively and dose-dependently impaired reversal learning [66]. Similar findings were reported in water maze tasks following the systemic or intra-mPFC blockade of the NMDA receptor GluN2B subunit with Ro25-6981 [67], which impairs reversal learning and increases the rate of perseverative errors [68]. In an appetitive plus-maze task requiring a switch between brightness and texture discrimination strategies, intra-mPFC MK-801 led to increased perseverative response and impaired shifting between discrimination strategies, whereas the intra-mPFC administration of the AMPA receptor antagonist LY293558 impaired acquisition [69].

Several studies examined the impact of genetically induced glutamatergic deficits on cognitive flexibility in spatial aversive tasks. Constitutive mutations affecting the NMDA NR1 subunit [70] or the AMPA GRIA1 subunit [71] lead to deficient acquisition in a spatial alternation task. The inactivation of the NR1 subunit limited to dorsal striatum neurons; however, had a specific impact on strategy shifting in the U-shaped water maze task, without affecting spatial acquisition or reversal learning [72].

Complementary evidence from appetitive paradigms reinforces the role of NMDA receptor function in cognitive flexibility, highlighting that the deficit may depend on the specific receptor subunit, brain region targeted, and task demands. For example, the constitutive deletion of *Grin2a* impaired set shifting but not acquisition or reversal in a Birrel–Brown-like task [73]. The homozygous deletion of cortical *Grin2b*, or OFC-specific *Grin2b* blocking by bilateral infusions of the selective GluN2B antagonist Ro 25-6981, resulted in specific reversal learning deficits in an operant conditioning task, without affecting acquisition [74].

While NMDA receptor blockers impair cognitive flexibility, NMDA receptor co-agonists—which increase glutamate transmission at the NMDA receptor—were shown to enhance it. For example, *DAO1^G181R^* mice, with a genetic inactivation of D-serine catabolic enzyme D-amino acid oxidase (DAO), exhibit increased D-serine levels and thus increased NMDA receptor activation. These mice show normal acquisition and substantially improved reversal learning in the MWM [75]. Interestingly, these mice also demonstrate an increased rate of extinction in the MWM, perhaps pointing to the enhanced reversal learning of reward contingencies. Similarly, the subcutaneous injection of D-serine (600 mg/kg; [68] to C57BL/6J mice did not alter acquisition but improved reversal learning. Taken together, these studies support the notion that the upregulation of NMDA receptor function facilitates cognitive flexibility.

Although most research on glutamate’s role in cognitive flexibility has centered on NMDA receptors, recent studies have also implicated other glutamate receptor subtypes, as well as presynaptic processes involved in glutamate synthesis, recycling, and release, as important contributors to this cognitive function. For example, the genetic ablation of mGluR5, a metabotropic glutamate receptor involved in presynaptic and postsynaptic aspects of glutamatergic transmission that affect both neurons and astrocytes [76,77], impairs reversal learning in an operant touchscreen-based task, leading to increased perseverative errors [78,79]. Conversely, increasing mGluR5 activity using positive allosteric modulators (PAMs) improved adaptive learning and reversal performance in the MWM [80] and rescued drug- or environmentally induced abnormalities in reversal learning [81,82].

On the presynaptic side, studies of vesicular glutamate transporters (VGLUTs) reveal that the heterozygous deletion of *Slc17a7*, mainly expressed in the cerebral cortex, hippocampus and cerebellar cortex [83,84], leads to reduced glutamate packaging and release along with selective reversal learning deficits in the MWM [85] and in an operant conditioning task [86].

Another pathway affecting synaptic glutamate levels involves the enzyme Glutamate Dehydrogenase 1 (Glud1), which catalyzes the deamination of glutamate into α-ketogluterate and is downregulated in the CA1 of patients with SZ [32]. Our group has shown that mice with CNS-specific reduction in Glud1 exhibit increased hippocampal glutamate levels and pyramidal neuron activity, along with cognitive flexibility deficits in a three-stage water T-maze task. In this task, mice first learn a spatial rule, then undergo reversal learning, and finally shift to finding the platform using a visual cue during the EDSS stage (Figure 1). Homozygous CNS-*Glud1*^−/−^ mice show impaired reversal and EDSS performance. Interestingly, heterozygous CNS-*Glud1*^+/−^ mice show intact reversal learning unless challenged with mild stress. Furthermore, performance in the reversal stage of this task correlates with the mPFC transcription of glutamate- and GABA-associated genes, meaning that the increased expression of glutamate markers and decreased expression of GABA markers are associated, with more trials required to complete the task [25,87].

Consistent with these findings, multiple studies suggest that disruptions in the excitatory/inhibitory (E/I) balance are linked to impaired cognitive flexibility. For example, a higher ratio of inhibitory to excitatory neurons (i.e., a reduced E/I balance) in the dentate gyrus has been associated with faster reversal learning, suggesting that enhanced local inhibition may support flexible updating [88]. In contrast, the optogenetic activation of glutamatergic neurons in the rat vmOFC resulted in deficient reversal learning in an operant conditioning paradigm, while the inhibition of glutamatergic firing improved reversal learning [89]. While these findings imply that excess glutamate release impairs flexibility, studies with VGLUT1-deficient mice (described above) indicate that diminished glutamate also impairs this cognitive capacity. This suggests that optimal levels of cortical and hippocampal glutamate are required for intact cognitive flexibility.

In SZ, higher hippocampal glutamate levels [90] and an abnormal mPFC glutamine/glutamate ratio [91] were correlated with compromised performance in the WCST. Notably, a recent study in healthy young adults did not find a correlation between striatal and ACC concentrations of glutamate/glutamine (Glx) levels and cognitive flexibility [92], so this association may be region- or disease-specific.

#### 3.3.2. GABA

Further support for the centrality of the E/I balance in cognitive flexibility comes from studies involving the manipulation of GABA transmission. Thus, the acute administration of TPA-023, a GABA_A_ partial agonist, significantly reversed PCP-induced deficits in reversal learning [93] in an operant reversal learning task. Studies in genetically modified mice point to OFC PV interneurons as particularly critical for reversal learning [94,95]. In support of the association between GABA transmission and reversal learning, parallel deficits in prefrontal GABA transmission and reversal learning in an operant task were found in mice heterozygous for a mutation in the *reelin* gene, an SZ risk gene [96]. Similar selective deficits in reversal learning were found in mice with a homozygous deletion of the cytoskeletal protein Radixin, which regulates the receptor density of the GABA_A_ receptors containing the α5-subunits [97]. However, blocking GABA_A_ receptors in the PFC left visual cue discrimination and reversal intact but impaired the shift to a new egocentric response strategy. Interestingly, the nature of the deficit depended on when GABA transmission was disrupted, producing either perseverative errors or novel, non-reinforced errors. Thus, PFC GABA disruption can induce ID or ED shift deficits, depending on the task and timing of GABA disruption [98]. In SZ, an MRS study revealed a negative correlation between PFC GABA levels and perseverative responding in patients with early-stage SZ [99], further supporting the importance of tight E/I balance regulation for this cognitive domain.

#### 3.3.3. Dopamine

There are five metabotropic dopamine receptors, classified according to structural and functional similarities. D1 and D5 receptors are structurally similar and primarily postsynaptic, typically exerting excitatory or modulatory effects. In contrast, D2, D3, and D4 receptors also share structural features but generally produce inhibitory effects and can be located on both pre- and postsynaptic sites [100]. Other components of the dopaminergic system include the dopamine transporter (DAT), encoded by the *Slc6a3* gene [101]. Dopaminergic signaling plays a key role in antipsychotic drug action [100] and has also been shown to modulate the E/I balance [102,103,104].

Dopamine involvement in cognitive flexibility has been investigated mainly in the context of non-spatial operant and digging tasks. Dopamine agonists, i.e., the dopamine releaser amphetamine, the D2/D3 receptor agonist quinpirole [105] and the selective D1-like agonist SKF81297 [106], disrupt reversal learning in lever pressing tasks. Amphetamine also disrupts reversal learning in the Birrel–Brown task, where its effects are reversed by D4 receptor antagonists L-745,870 [107]. Curiously, manipulations that hinder dopamine receptor function also disrupt flexibility. For example, chronic treatment with the typical antipsychotic D2 receptor blocker haloperidol impairs reversal and set shifting [108]. Likewise, mice lacking the D2 receptor exhibit impairments in olfactory reversal learning [109,110] and the knockout of the D2 long receptor subtype impairs both acquisition and reversal in a visual discrimination task [111]. The selective knockout of D2, but not D1, receptors from GABA interneurons also selectively impairs reversal. However, the transient overexpression of D2 receptors in the striatum leads to selective deficits in reversal learning in olfactory reversal learning task, leaving acquisition unaffected [112]. In conclusion, as with glutamate and GABA signaling, it seems that cognitive flexibility is enabled under conditions of optimal dopamine transmission. In SZ patients, dopamine D2/3 receptors’ binding affinity in the caudate nucleus and putamen correlates with TMT-B performance [113]. While all currently prescribed antipsychotic drugs target dopamine receptors, their impact on cognitive flexibility in patients with SZ is unclear and may depend on genetic predisposition [114].

#### 3.3.4. Serotonin

The serotonergic system consists of seven receptor subtypes (5-HT1 to 7), located pre- or post-synaptically, which can inhibit (5-HT1B or 5-HT1A, respectively) or potentiate transmission (e.g., 5-HT2A) [115]. The serotonin transporter 5-HTT is encoded by the *Slc6a4* (SERT) gene [116]. Importantly, as with dopamine, there is evidence to suggest that serotonergic signaling can modulate the E/I balance [117,118].

As with dopamine, the involvement of serotonin in cognitive flexibility was mainly investigated using appetitive tasks. For example, pharmacological blockade or genetic deletion (either partial or complete) of the serotonin transporter 5-HTT resulted in fewer errors in reversal learning in a lever pressing task [119]. Interestingly, the selective 5-HTT inhibitor fluoxetine, commonly used as an antidepressant, improved performance in an operant learning task, specifically reducing the rate of impulsive and perseverative errors [120].

The complexity of the serotonergic system is reflected in the various contrasting findings. For example, the systemic administration of either a 5-HT2A agonist [121] or antagonist [122] impaired reversal learning in appetitive tasks. Interestingly, different receptors, e.g., 5-HT2A and 5-HT2C may be associated with different error types, i.e., regressive or perseverative, and have opposite effects on flexibility measures [121]. In other studies, both systemic [122] or OFC-specific [123] 5-HT2C receptor blockade improved reversal learning.

In sum, the serotonergic system plays a role in cognitive flexibility, but its precise contribution is obstructed by the variability in behavioral tasks and the lack of a methodological approach regarding different manipulations affecting serotonin synaptic levels. In SZ, patients with polymorphisms in the genes encoding the 5-HT2A receptor [124], or the serotonin transporter 5-HTT [125], demonstrated significantly worse performance in the WCST.

#### 3.3.5. Additional Neural Substrates

While this review synthesizes the most consistent patterns observed across studies, alternative findings implicating different neurotransmitter systems, neural circuits, and genes have also been reported. For example, cognitive flexibility was found to be altered in mice with genetic modifications to genes encoding the membrane channel protein *Panx1* [126], the serine/threonine kinase *Mapkapk2* [127], and the GTPases *Rab3a* [128] and *Rab3B* [129], which are involved in vesicle fusion and release. These genes control neuronal function but do not specifically modulate the E/I balance or dopamine release and are also associated with a wide range of synaptic and behavioral alterations. Cognitive flexibility was also found to be modulated by the endocannabinoid system [130,131], the acetyl cholinergic system [132,133] and by sex hormones such as estrogen and its receptors [134,135].

#### 3.3.6. Environmental Manipulations

Environmental manipulations, particularly in social stress, affect reversal learning and EDSS. Han et al. [136] showed that a 2-week social isolation period in juvenile rats (PND 21–34) resulted in selectively impaired reversal learning without affecting acquisition in the MWM, and our group [137] similarly found that a 3-week social isolation period starting in mid-adolescence (PND 38), but not in adulthood (PND 60), impaired reversal and EDSS in the water T-maze while increasing the expression of glutamate markers in the mPFC.

Non-social stress was also investigated in the context of cognitive flexibility and was found to have detrimental effects. Butts et al. [138] and Thai et al. [139] both found that acute stress impairs cognitive flexibility, but the affected stage differed depending on task design and timing. In both studies, stress impacted the final stage of testing—set shifting in Butts et al. and reversal in Thai et al.—suggesting that task order and experimental design (between- vs. within-subjects) may influence the observed effects of stress. Differences in methodology likely account for the variation in performance outcomes between the two studies.

Early life stress led to selective deficits in reversal learning in adult mice and decreased PV and GAD67 in OFC. Interestingly, the behavioral effect and changes in GAD67 were found in female mice only and were mimicked by the optogenetic silencing of PV cells in OFC, but not in the mPFC [140]. This suggests that certain environmental factors may influence cognitive flexibility by altering the E/I balance.

Gestational manipulations, such as treatment with the viral mimic PolyI:C, impair cognitive flexibility in adult animals. For example, maternal PolyI:C treatment in rodents resulted in impaired reversal in offspring [141,142]. The effect of this manipulation has been attributed to the activation of the immune system during gestation, which impacts the development of the hippocampus and mPFC circuitry [143,144] and dopamine neurotransmission [145].

Some environmental changes can enhance performance in cognitive flexibility tasks. For example, environmental enrichment improved reversal learning in operant tasks [146] [147]. Environmental enrichment also improved reversal learning in a four-radial arm water maze and led to higher c-Fos expression in the mPFC (specifically the cingulate cortex) and the OFC [148]. In some, some changes to the environment—some of which, e.g., immune activation during pregnancy [149] and social stress [150], have been associated with SZ psychopathology—lead to perseverative behavior in cognitive flexibility task, while other manipulations can have a beneficial impact on this cognitive capacity.

**Table 1 biomolecules-15-01154-t001:** Detailed overview of animal model findings (2005–2025) summarizing valence, task demands and neural substrates of reversal and/or attentional set shifting deficits.

References	Mice/Rats	Aversive/Appetitive	Manipulation	Methods	Task	Reversal	EDSS	Brain Region	Neurotransmitters
Bardgett et al., 2003 [59]	Mice	Aversive	Pharmacological	Acute injection of MK-801	water T-maze	↓			Glutamate
Chadman et al., 2006 [60]	Rats	Appetitive	Pharmacological	Acute injection of MK-801	water T-maze	↓			Glutamate
McLean et al., 2010 [61]	Rats	Appetitive	Pharmacological	Acute injection of PCP	Operant conditioning	↓			Glutamate
Abdul-Monim et al., 2006 [62]	Rats	Appetitive	Pharmacological	Acute injection of PCP	Operant conditioning	↓			Glutamate
Patrono et al., 2023 [63]	Rats	Appetitive	Optogenetic stimulations	Injection of MK-801	ASST	↓, ↑	↓, ↑	PFC, ventral hippocampus	GABA
Li et al., 2016 [64]	Rats	Aversive	Pharmacological	Chromic injection of MK-801	Morris water maze	↓			Glutamate
Thonnard et al., 2019 [65]	Rats	Aversive	Pharmacological	Chromic injection of MK-801	Morris water maze	↓			Glutamate
Watson & Stanton, 2009 [66]	Rats	Appetitive	Pharmacological	bilateral intrahippocampal administration of MK-801	T-maze	↓			Glutamate
Dong et al., 2013 [67]	Rats	Aversive	Pharmacological	systemic or intra-hippocampal blockade of NMDA receptor *Grin2b* subunit	MWM	↓		Hippocampus	Glutamate
Duffy et al., 2008 [68]	Mice	Aversive	Pharmacological	blockade of NMDA receptor *Grin2b* subunit, D-Serine administration	MWM	↓, ↑			Glutamate
Darvas & Palmiter, 2015 [72]	Mice	Aversive	Genetic	*Grin1* knockout	Water U-maze	-	↑	Dorsal striatum	Dopamine
Marquardt et al., 2014 [73]	Mice	Appetitive	Genetic	*Grin2a* knockout	ASST	-	↓		Glutamate
Brigman et al., 2013 [74]	Mice	Appetitive	Genetic and Pharmacological	Cortical *Grin2b* knockout and OFC-specific *Grin2b* blocking	Operant conditioning	↓		mPFC and OFC	Glutamate
Labrie et al., 2009 [75]	Mice	Aversive	Genetic	DAO1^G181R^ mice with inactivation of DAO enzyme	MWM	↑			Glutamate
Zeleznikow-Johnston et al., 2018 [78]	Mice	Appetitive	Genetic	*mGluR5* knockout	Operant conditioning	↓			Glutamate
Lim et al., 2019 [79]	Mice	Appetitive	Genetic	*mGluR5* knockout	Operant conditioning	↓			Glutamate
Xu et al., 2013 [80]	Mice	Aversive	Pharmacological	increasing *mGluR5* activity using positive allosteric modulators	MWM	↑			Glutamate
Gastambide et al., 2012 [81]	Rats	Appetitive	Pharmacological	increasing *mGluR5* activity using positive allosteric modulators	ASST	↓, ↑	↓		Glutamate
Joffe et al., 2019 [82]	Mice	Appetitive	Pharmacological	increasing *mGluR5* activity using positive allosteric modulators	Operant conditioning	↑			Glutamate
Balschun et al., 2010 [85]	Mice	Aversive	Genetic	*Vglut1* knockout	MWM	↓			Glutamate
Granseth et al., 2015 [86]	Mice	Appetitive	Genetic	*Vglut1* knockout	Operant conditioning	↓			Glutamate
Lander et al., 2019 [32]	Mice	Aversive	Genetic	Knockdown and knockout of *Glud1*	Water T-maze	↓	↓		Glutamate
Asraf et al., 2023 [25]	Mice	Aversive	Genetic	Knockdown of *Glud1*	Water T-maze	↓		mPFC	Glutamate
Morellini et al., 2010 [88]	Mice	Aversive	Genetic	Knockout of *Tnr*	MWM	↑		Hippocampus	Glutamate and GABA
Barnes et al., 2023 [89]	Rats	Aversive	Optogenetic stimulations	Optogenetic activation orinhibition of glutamatergic neurons in vmOFC	Operant conditioning	↓, ↑		OFC	Glutamate
Rajagopal et al., 2018 [93]	Mice	Appetitive	Pharmacological	Administration of TPA-023, a GABAA partial agonist	Operant conditioning	↓, ↑			Glutamate and GABA
Brigman et al., 2006 [96]	Mice	Appetitive	Genetic	Knockdown of *reelin*	Operant conditioning	↓	-	PFC	GABA
Hausrat et al., 2015 [97]	Mice	Aversive	Genetic	Knockout of *Rdx*	MWM	↓			GABA
Bissonette et al., 2010 [94]	Mice	Appetitive	Genetic	Knockout of *Plaur*	Foraging reversal	↓		OFC	GABA
Bissonette et al., 2015 [95]	Mice	Appetitive	Genetic	Knockout of *Plaur*	Foraging reversal	↓		OFC	GABA
Boulougouris et al., 2009 [105]	Rats	Appetitive	Pharmacological	Administration of D2/D3 receptor agonist	Operant conditioning	↓			Dopamine
Izquierdo et al., 2006 [106]	Mice	Appetitive	Pharmacological	Administration of D1-like agonist	Operant conditioning	↓			Dopamine
Connolly et al., 2014 [107]	Rats	Appetitive	Pharmacological	Administration of D4 receptor antagonist	ASST	↓, ↑	-		Glutamate and Dopamine
DeSteno & Schmauss, 2009 [108]	Mice	Appetitive	Pharmacological	Administration of typical antipsychotic D2 receptor blocker haloperidol	ASST	↓	↓		Dopamine
Kruzich & Grandy, 2004 [109]	Mice	Appetitive	Genetic	Knockout of D2 receptor	Odor discrimination	↓			Dopamine
Kruzich et al., 2006 [110]	Mice	Appetitive	Genetic	Knockout of D2 receptor	Odor discrimination	↓			Dopamine
Morita et al., 2016 [111]	Mice	Appetitive	Genetic	Knockout of D2 long receptor	Operant conditioning	↓			Dopamine
Kellendonk et al., 2006 [112]	Mice	Appetitive	Genetic	Transient overexpression of D2 receptors in striatum	Odor discrimination	↓			Dopamine
Brigman et al., 2010 [119]	Mice	Appetitive	Genetic and Pharmacological	Pharmacological blockade or genetic deletion (either partial or complete) of serotonin transporter 5-HTT	Operant conditioning	↑			Serotonin
Odland et al., 2021 [120]	Mice	Appetitive	Pharmacological	Administration of selective 5-HTT inhibitor fluoxetine	Operant conditioning	↑			Serotonin
Amodeo et al., 2020 [121]	Mice	Appetitive	Pharmacological	Administration of 5-HT2A agonist and/or 5-HT2C antagonist	T-maze	↓			Serotonin
Boulougouris et al., 2008 [122]	Rats	Appetitive	Pharmacological	Administration of 5-HT2A or 5-HT2C antagonists	Operant conditioning	↓, ↑			Serotonin
Boulougouris et al., 2010 [123]	Rats	Appetitive	Pharmacological	Administration of Intra-OFC 5-HT2C receptor antagonism	Operant conditioning	↑		OFC	Serotonin
Han et al., 2011 [136]	Rats	Aversive	Environmental	Social isolation	MWM	↓			
Lander et al., 2017 [137]	Mice	Aversive	Environmental	Social isolation	Water T-maze	↓	↓	PFC	
Butts et al., 2013 [138]	Rats	Appetitive	Environmental	Stress	Operant conditioning	-	↓		
Thai et al., 2013 [139]	Rats	Appetitive	Environmental	Stress	Operant conditioning	↓	-		
Goodwill et al., 2018 [140]	Mice	Appetitive	Environmental	Stress	ASST	↓	↓	OFC and mPFC	GABA
Zeleznikow-Johnston et al., 2017 [146]	Mice	Appetitive	Environmental	Environmental enrichment	Operant conditioning	↑			
Kikuchi et al., 2022 [147]	Mice	Appetitive	Environmental	Environmental enrichment	Operant conditioning	↑			
Sampedro-Piquero et al., 2015 [148]	Mice	Aversive	Environmental	Environmental enrichment	4-radial arm water maze	↑		OFC and mPFC	
Aarde et al., 2021 [151]	Mice	Appetitive	Environmental	Sex differences	Operant conditioning	↓		OFC and mPFC	

↓: deficient performance. ↑: improved performance. -: no change in performance. Gray shading: not assessed.

## 4. Summary

This review summarizes current research on cognitive flexibility, focusing particularly on insight emerging from rodent models. We examined reversal learning and set shifting across both aversive and appetitive paradigms, recognizing that while these paradigms may assess comparable cognitive processes, they engage distinct neural circuits. As a result, findings across paradigms may appear inconsistent or contradictory.

Reversal learning: Aversive reversal tasks, typically based on spatial paradigms, primarily engage hippocampal–medial prefrontal cortex (mPFC) circuitry. In contrast, appetitive reversal tasks—often involving non-spatial cues—rely heavily on the orbitofrontal cortex (OFC), which is key for adjusting to changes in reward contingencies. Although the mPFC contributes to both task types, the OFC appears especially crucial in appetitive contexts. Additionally, the striatum is frequently implicated in appetitive but not aversive reversal tasks. Across both paradigms, glutamate and GABA signaling in the PFC are essential for successful reversal learning. In appetitive tasks, serotonergic and dopaminergic systems, especially within the striatum, also play a prominent role.

EDSS: In aversive spatial tasks, EDSS typically involves switching from a spatial to a visual strategy and depends on an intact mPFC. In appetitive paradigms, EDSS requires shifting attention between distinct sensory modalities (e.g., olfactory to tactile), engaging a broader network including the mPFC, ventral hippocampus, and anterior cingulate cortex (ACC). As with reversal learning, glutamatergic and GABAergic transmission support EDSS across both contexts, though serotonergic and noradrenergic pathways have been specifically implicated in appetitive EDSS tasks.

Notably, the behavioral distinction between reversal learning and EDSS is not always clear, leading to overlap in brain circuits involved in these learning tasks. For example, reversal learning is sometimes performed after EDSS [38]; neural activation during pre-EDSS and post-EDSS reversal may differ substantially.

Several points should be made regarding the interpretation of findings in aversive and appetitive paradigms. First, neural substrates that appear to differ between aversive and appetitive tasks may be due to inherent differences between these two types of paradigms. However, these differences in circuitry may partly be due to conventions in the field of learning. For example, the striatum is not commonly assessed in spatial aversive EDSS tasks whereas hippocampal–mPFC circuitry is often examined, and the hippocampus may be overlooked in appetitive-based paradigms.

A second important consideration is whether the observed deficits truly reflect impairments in the targeted cognitive domain—such as reversal learning or attentional set shifting—or are instead due to increased task difficulty or memory demands. For example, when EDSS deficits are found in a water T-maze task, these deficits may not be due to compromised extra-dimensional shifting but rather to the fact that this stage is simply more challenging than the previous reversal stage. Alternatively, apparent difficulties in attentional set shifting may reflect sensory limitations of a particular modality. For example, rodents are poor visual discriminators, and compromised performance in visual discrimination tasks may be modality-specific. An alternative explanation for apparent cognitive flexibility deficits is changes in motivational state; for example, dopamine manipulations may influence motivation—especially in reward-based tasks—potentially leading to the misinterpretation of selective reversal learning deficits. A possible way to address these concerns is to control for task difficulty by counterbalancing modalities or by testing rodents in additional tasks with increasing task difficulty or using different motivational states but no flexibility demands.

Third, most rodent studies—with few exceptions—were performed in males, despite evidence of sex differences in cognitive flexibility abilities in human studies [152]. A mouse study that examined sex differences found that while male and female mice do not differ in the number of trials to reach criterion in the reversal stage of a cognitive flexibility task, the type of errors made are sex-dependent, with males making more perseverative errors than females [151]. This suggests that the neural mechanisms supporting cognitive flexibility may differ between sexes, with potential implications for both the diagnosis and treatment of flexibility-related impairments in humans. Furthermore, accumulating evidence shows that sex differences exist in brain structure [153,154], hormone regulation [155], reward and addiction [156], immune activity [157], stress reactivity [158,159] and behavioral strategies [160,161], all of which can impact cognitive flexibility. In SZ, symptoms manifest differently in males and females [162]. Thus, relying solely on one sex, typically males, limits the generalizability of findings and overlooks biologically relevant variability, with potential impact on diagnosis and treatment.

Relatedly, many of the reviewed rodent studies assessed the number of trials required to reach criterion or the time to respond but did not evaluate error types, e.g., perseverative vs. regressive errors. Different error types may reflect different neural circuitry affected by disease processes or by specific manipulations and should be assessed along with commonly used measures such as error rate or trials to criterion.

## 5. Conclusions

Cognitive flexibility is a core executive function that relies on the integrity of hippocampal–prefrontal circuitry and proper striatal function. Future studies should consider the motivational valence of the task as it can significantly influence the neural circuits involved. Appetitive- and aversive-driven cognitive flexibility deficits may reflect different pathological processes that are impaired in SZ, as well as other psychiatric illnesses. Moreover, valence differences in cognitive inflexibility may characterize different stages of SZ; for example, a positive-symptom-based stage of illness may be characterized by ‘appetitive’ (reward-based) cognitive inflexibility, whereas chronic schizophrenia, associated with negative symptoms, may be characterized by a more ‘aversive’ cognitive inflexibility profile. This distinction could also guide therapeutic or pharmacological intervention; patients with a positive-valence (appetitive) cognitive flexibility deficit could benefit more from therapeutics that target striatal dopamine, whereas cognitive flexibility deficits in aversive settings may benefit from pharmacology targeting the prefrontal E/I balance. Thus, a better understanding of the contribution of valence to cognitive flexibility may contribute to diagnosis and treatment.

Animal and human studies should also control for task difficulty and potential sex differences. A translational framework that bridges findings across species may improve the diagnosis and treatment of cognitive impairment in SZ and other conditions characterized by deficits in cognitive flexibility.

## Figures and Tables

**Figure 1 biomolecules-15-01154-f001:**
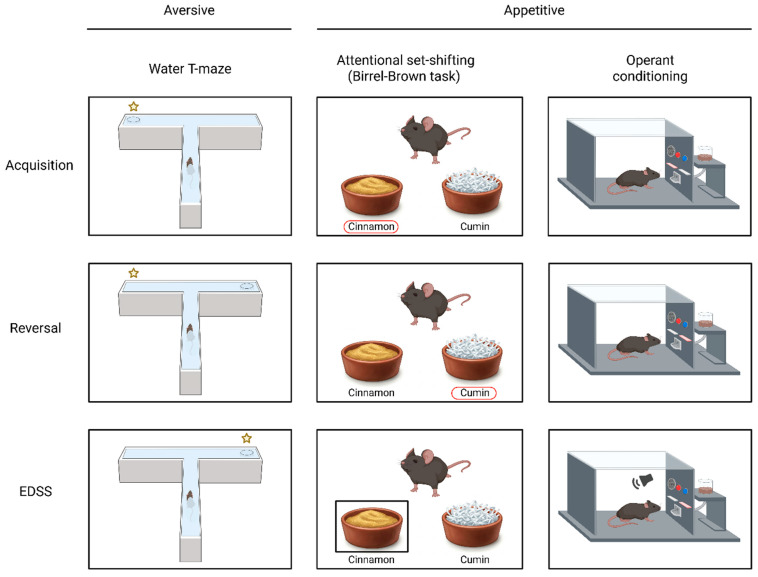
Tasks commonly used to assess cognitive flexibility in rodents. (**Left**): Water T-maze tasks. Rodents are required to learn the spatial location of a platform (top), which is then reversed (middle), and finally have to locate the platform according to a visual cue (bottom). Yellow star: correct response. (**Middle**): The attentional set-shifting (Birrel–Brown) task. Rodents are required to locate a food award according to odor A (top), and after reaching criterion, the rewarded odor is reversed (middle). In the EDSS phase (bottom), rodents have to locate the food reward according to the digging media (right). Red or black frame: correct response. (**Right**): Operant learning tasks. Rodents are trained to acquire an operant response (lever press) to a (visual) cue (top). The rewarded cue is then reversed (middle). In the EDSS phase, the reward is obtained when the animal responds to a cue from a different dimension which was not previously rewarded (bottom). Red bar: correct response.

## Data Availability

No new data were created or analyzed in this study.

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
