# Peer review of "Valence-Driven Cognitive Flexibility: Neurochemical and Circuit-Level Insights from Animal Models and Their Relevance to Schizophrenia"

_biomolecules, 2025, doi:10.3390/biom15081154_

Round 1
Reviewer 1 Report
Comments and Suggestions for Authors
The review paper by Asraf and Gaisler-Salomon addresses cognitive flexibility and the valence driven context to elaborate on the role of brain regions and key neurotransmitter systems. They report effects of aversive versus appetitive valence in humans and animal models in cognitive flexibility tests, leading to conceive distinct neurobiological mechanisms. They report data on the neurobiological substrates that underline each type of valence, sometimes overlapping, and sometimes not. They finally address the role of neurotransmitter systems with a focus on glutamate, GABA, noradrenaline, dopamine, and serotonin, and environmental manipulation.
The article is well written and is logically developed. The core idea (valence driven cognitive flexibility) is interesting but I feel that the authors narrowed their analysis to some brain regions, and some tests. The distinction of brain regions and neurotransmitter systems in ID/EDSS should be supported by additional data. The idea of the table 1 could be developed for each neurotransmitter. If it is not possible, or difficult, then, I’m wondering what is the meaning of the table 1 which could be really misleading.
There are numerous regions engaged in responses corresponding to cognitive flexibility. The authors should explain better their intent in looking at 4/5 regions, because valence has its neurobiological substrates involving also other brain regions.
The part on neurotransmitter is less convincing, again not supported by a sufficient corpus of data.
Minor
Line 348: schizophrenia should be SZ
Line 319: repetition on “in”
Line 253: consider to break down the sentence (too long with two ideas).
Line 152: in instead of is
Line 83: suggesting instead of indicating.
Reviewer 2 Report
Comments and Suggestions for Authors
In the review entitled “Valence-Driven Cognitive Flexibility: Neurochemical and Circuit-Level Insights from Animal Models and Their Relevance to Schizophrenia” the authors review cognitive flexibility and its dysfunction in schizophrenia. The article presents a novel approach to motivational valence in interpreting behavioral tasks. The review integrates recent studies with a well-detailed review of circuits, neurotransmitters, and environmental manipulation.
There are a few points that can be addressed.
- Describe the methodological approach of the review, especially specifying the search criteria and the selection of studies. This does not require a PRISMA protocol.
- The review could expand on how these valential differences could guide therapeutic or pharmacological interventions in schizophrenia.
- Expand on sex biases in animal models.
4. More attention could be given to humans, where neuroimaging or genetic findings would be deeply integrated, for example with respect to glutamate or dopamine.
Round 2
Reviewer 1 Report
Comments and Suggestions for Authors
Dear authors,
It is indeed better to indicate the way it was done. However, there is still the problem of the table 1 that has not been changed with respect to the previous version. You might have forgotten to replace the old one.
Author Response
comments:
They did not change the table 1 even though they indicated that they did it in their response. I believe they forgot, so it should be fast.
response:
Thank you for your comment. There may be something wrong with the submission system or the website. I am attaching the revised version again
